# Mechanical Properties and Wear Resistance of Commercial Stainless Steel Used in Dental Instruments

**DOI:** 10.3390/ma14040827

**Published:** 2021-02-09

**Authors:** Hye-Bin Go, Jae-Yun Bang, Kyoung-Nam Kim, Kwang-Mahn Kim, Jae-Sung Kwon

**Affiliations:** 1Department and Research Institute of Dental Biomaterials and Bioengineering, Yonsei University College of Dentistry, Seoul 03722, Korea; hbgo@yuhs.ac (H.-B.G.); kmkim@yuhs.ac (K.-M.K.); 2BK21 FOUR Project, Yonsei University College of Dentistry, Seoul 03722, Korea; 3Department of Dental Hygiene, Kyungdong University, Wonju 26495, Gangwon-do, Korea; mare190@naver.com (J.-Y.B.); kimkndds@kduniv.ac.kr (K.-N.K.)

**Keywords:** dental instrument, ultrasonic scaler tip, stainless steel, Vickers hardness, wear resistance

## Abstract

The aim of this study was to investigate the element composition and grain size of commercial dental instruments used for ultrasonic scaler tips, which are composed of stainless-steel materials. The differences in mechanical properties and wear resistances were compared. The samples were classified into 4 groups in accordance with the manufacturer, Electro Medical Systems, 3A MEDES, DMETEC and OSUNG MND, and the element compositions of each stainless-steel ultrasonic scaler tip were analyzed with micro-X-ray fluorescence spectrometry (μXRF) and field-emission scanning electron microscopy (FE-SEM) with energy-dispersive X-ray spectroscopy (EDS). One-way ANOVA showed that there were significant differences in shear strength and Vickers hardness among the stainless-steel ultrasonic scaler tips depending on the manufacturer (*p* < 0.05). The mass before and after wear were found to have no significant difference among groups (*p* > 0.05), but there was a significant difference in the wear volume loss (*p* < 0.05). The results were then correlated with μXRF results as well as observations of grain size with optical microscopy, which concluded that the Fe content and the grain size of the stainless steel have significant impacts on strength. Additionally, stainless-steel ultrasonic scaler tips with higher Vickers hardness values showed greater wear resistance, which would be an important wear characteristic for clinicians to check.

## 1. Introduction

The lack of daily tooth brushing after ingesting food leads to plaque formation at the gingival area of the tooth surface, and the plaque becomes mineralized over time, resulting in dental calculus [1]. Increased plaque and calculus cause periodontal disease by the risk factor for infection [2,3,4] and can cause halitosis and pain if plaque and calculus are not managed. Also, such insufficient oral hygiene can lead to serious conditions of poor oral health in implant failure. It can induce serious complications during dental implants placement, especially in patients infected with human immune virus [4,5]. Therefore, for good and healthy oral hygiene, various manual or powered devices (scaling and root planning) that can remove dental plaque and calculus from tooth-crown and root surfaces must be used on a regular basis [6]. In recent years, as the paradigm has shifted from treatment-oriented dental care to prevention-oriented dental care, the importance of scaling as a preventive treatment has been increasingly emphasized. Scaling is used to debride the dental plaque and calculus of the tooth crown and root surfaces during routine periodontal treatment [7].

The instruments currently used for scaling can be classified into two types: hand scalers and ultrasonic scalers. Both types of scalers are available, but compared to hand instruments, ultrasonic scalers have been widely used in recent years because they require less time for scaling, increase working efficiency and offer ergonomic benefits and simplicity of use to clinicians [8,9,10]. Ultrasonic scalers remove deposits on the tooth surface, such as bacterial plaque, calculus and endotoxins, by utilizing vibrations of approximately 24,000 to 42,000 Hz with cooling water [7,11]. Ultrasonic scaler tips are made of various materials, such as stainless steel, copper and plastic [12,13]. The efficiency of scaler tips varies depending on the manufactured material; stainless steel tips are two times more efficient than copper tips, and copper tips are twenty times more efficient than plastic tips. In addition, the hardness value of stainless steel is 610 HV, which is higher than that of copper (89 HV) and plastic (37 HV) [12]. Copper and plastic tips have low hardness, so they are easier to wear than stainless steel tips. When the tip is worn, the vibration of the scaler tip is reduced, which reduces the efficiency of tartar removal [14]. Tip wear of 1 mm results in a loss of efficiency of approximately 25%, and wear of 2 mm results in a loss of efficiency of approximately 50% [15]. For this reason, stainless steel ultrasonic scaler tips have generally been used the most [12,13].

The requirements for scaler tips according to ISO 18397 include torque retention, insertion of the scaler tip, frequency, supply of cooling liquid, noise level, etc. [16]. However, these requirements are for the combination and operation of the scaler handpiece and the scaler tip and are not requirements for the scaler tip itself, such as the mechanical and physical properties of the scaler tip. Previous research on scaler tips also studied the material of the scaler tips, roughness due to wear, etc., instead of the requirements of ISO 18397. Bruna et al. observed the surface roughness of titanium discs using a stainless-steel tip and a copper alloy silver-plated tip [17], and Seung-Ho et al. used atomic force microscope (AFM) to research stainless steel tips and copper tips, plastic tip surface roughness and efficiency [12]. These studies observed that the surface of the Ti disc changed with the scaler tip and did not observe the surface of the worn scaler tip. Simon et al. confirmed the change in amplitude due to wear by artificially removing 2 mm of the tip length of a magnetostrictive ultrasonic device [14]. However, there are no studies investigating the element composition and grain size of commercial stainless steel ultrasonic scaler tips and to compare the differences in mechanical properties and wear resistance according to this.

Therefore, the aim of this study was to investigate the element composition and grain size of commercial stainless steel ultrasonic scaler tips and to compare the differences in mechanical properties and wear resistance according to this. The null hypotheses were as follows: (1) there would be no difference in mechanical properties and wear resistance depending on the element composition of the commercial stainless steel ultrasonic scaler tips (2) there would be no difference in mechanical properties and wear resistance depending on the grain size of the commercial stainless steel ultrasonic scaler tips.

## 2. Materials and Methods 

### 2.1. Sample Preparation

Four commercial stainless steel ultrasonic scaler tips were selected as tips from various manufacturers widely used in clinics (Figure 1). The manufacturer information and specifications of the selected stainless steel ultrasonic scaler tips are described in Table 1. The samples were divided into four groups of 26 samples each according to the manufacturer.

### 2.2. Elemental Composition Analysis

The elemental compositions of the stainless-steel ultrasonic scaler tips were determined using micro-X-ray fluorescence spectrometry (μXRF) analysis (M4 TORNADO, Bruker, Billerica, MA, USA) with a rhodium filament operating at 50 kV and 40 mA and with a spot size of 25 µm. The system comprises a Rh anode X-ray tube and an XFlash detector (energy resolution 145 eV for Mn-Kα and 30 mm^2^ active). Two sites were measured at random for each sample, and the mean values were obtained and compared. Also, the field-emission scanning electron microscopy (FE-SEM; MERLIN, ZEISS, Ltd., Oberkochen, Germany) equipped with energy-dispersive X-ray spectroscopy (EDS) were used to determine the elemental compositions.

### 2.3. Shear Strength Test

The shear strength of seven samples per test group was measured. Before the strength measurement, 2 mm inward from the end of the working end of all the samples was measured and marked. The marked position of the samples was inserted between metal blocks and fixed firmly. Then, the metal block was fixed by aligning the jig of a universal testing machine (Proline, Zwick, Ulm, Germany) parallel to the contact area of the metal block (Figure 2). The test was performed with the chisel-shaped jig of the universal testing machine, and the crosshead speed was 1.5 mm/min. The shear strength value was calculated by dividing the breaking load F (N) by the bonded area (mm^2^). At this time, the area of the stainless-steel ultrasonic scaler tip cut 2 mm from the tip end was measured to calculate the bonded area.

### 2.4. Vickers Hardness Test

Ten samples were prepared for each stainless-steel ultrasonic scaler tip group. When measuring the Vickers hardness of the scalar tip, the end of the scalar tip was fixed flat with the bottom, and was measured at a surface of 2 mm at the end of the tip. The Vickers hardness was determined with a Vickers hardness tester (DMH-2, Matsuzawa Seiki Co., Tokyo, Japan) using a Vickers diamond indenter and a 500 g load applied for 10 s, and both indentation diagonals (mm) were measured. Vickers hardness (HV) was determined using the following equation:(1)$$HV\;=\;1.854\;\times \;\frac{F}{d^{2}}$$
where *F* is the indentation load (kgf), and *d* is the arithmetic mean of two diagonals (mm). The samples were randomly measured three times each at a site within end 3 mm of the stainless-steel ultrasonic scaler tip.

### 2.5. Wear Test

The wear test of eight samples per test group was measured by using ceramic blocks and a portable ultrasonic scaler generator (Ultrasonic scaler, 3M ESPE, Seoul, Korea). The size of the ceramic block was (12.0 ± 0.2) × (4.0 ± 0.2) × (3.0 ± 0.2) mm, and one ceramic block was used per group. To obtain a consistent angle for all stainless-steel ultrasonic scaler tips, a custom jig that can fix the ultrasonic scaler handpiece and the ceramic block was fabricated (Figure 3A). The ultrasonic scaler handpiece could be fixed through the screws on the left and right sides, and the ceramic block could be fixed with one screw. As the first step, the ultrasonic scaler handpiece and ceramic block were placed in the jig. At this time, the ceramic block was fixed to be horizontal to the floor, and the ultrasonic scaler handpiece was fixed so that the long axis of the stainless-steel ultrasonic scaler tip working end was perpendicular to the floor (Figure 3A). The stainless-steel ultrasonic scaler tip was placed perpendicular to the ceramic block and positioned at the center of the ceramic block with a continuous force of approximately 310 g (combined weight of the ceramic block and fixture). After confirming that each fixture was in the correct position when viewed from below, the portable ultrasonic scaler generator was operated for 60 min per sample (Figure 3B). The portable ultrasonic scaler was set to a power of 32 kHz (maximum) and water injection rate of 60 mL/min (maximum).

### 2.6. Wear Degree by Mass and Volume for Stainless-Steel Ultrasonic Scaler Tips

Two methods of calculating the wear by mass and volume were used in this study. To evaluate the mass change due to the wear test, the mass of the stainless-steel ultrasonic scaler tip was measured using an analytical balance (Sartorius analytic A120S, Sartorius, Göttingen, Germany). The wear degree by mass of all samples was calculated by measuring the mass before and after the wear test.

To evaluate the wear volume loss, the amount of wear of the stainless-steel ultrasonic scaler tip was calculated. Since the stainless-steel ultrasonic scaler tip wears in the shape of a semi-ellipsoid, the wear degree was measured through the ellipsoid volume formula. The wear degree was calculated by the following equation:(2)$$W\;=\;\frac{2}{3}\pi HFL$$
where W is the wear, *H* is the height of the semi-ellipsoid (mm), *F* is radius 1 of the ellipse, and *L* is radius 2 of the ellipse (Figure 4). All lengths used for calculating the wear volume loss of stainless-steel ultrasonic scaler tips were measured using a profile projector (PJ-A3000, Mitutoyo, Takatsu-ku, Japan). The height of the semi-ellipsoid (H) was obtained via the difference between the length of the stainless-steel ultrasonic scaler tip measured before the wear test and the length of the stainless-steel ultrasonic scaler tip measured after the wear test. At this time, a line was drawn by connecting the points 0.5 mm and 1.0 mm from the working end of the stainless-steel ultrasonic scaler tip, placing the line parallel to the Y axis of the profile projector, and then measuring the line. The measurement was made from the end of the working end of the stainless-steel ultrasonic scaler tip to the start of the shank of the stainless-steel ultrasonic scaler tip. Ellipse radius 1 (F) and ellipse radius 2 (L) were obtained by measuring the lengths of the inner side and lateral side of each stainless-steel ultrasonic scaler tip after the wear test.

The degree of wear on the stainless-steel ultrasonic scaler tip was visually checked on the sides of the stainless-steel ultrasonic scaler tip using an image analyzer (Micro Hiscope System, ING PLUS, Seoul, Korea). At this time, the images of all samples were acquired in the same position for comparison before and after wear of the stainless-steel ultrasonic scaler tip. Then, the degree of frontal wear of the stainless-steel ultrasonic scaler tip was measured with field emission scanning electron microscope (FE-SEM; MERLIN, ZEISS, Ltd., Oberkochen, Germany) before and after the wear test. The samples were platinum-coated using an ion coater (ACE600, Leica, Ltd., Sidney, Austria).

### 2.7. Grain Size

The microstructure surface samples were mounted on epoxy resin so that the working end of the scaler tip was parallel to the floor, and then mechanically polished sequentially using 800-grit, 1000-grit, 1500-grit, 2000-grit SiC papers and standard colloidal silica suspension. Then, the well-polished samples were etched for 2–3 min using a solution of 30 mL hydrochloric acid, 30 mL nitric acid and 40 mL distilled water to reveal the grain boundaries [18]. After that, optical images at 500× magnification and 1000× magnification were captured with an optical microscope (OM, JE clipse LV150N, Nikon, Tokyo, Japan). The average grain size was determined based on the OM images using ImageJ software (1.53v) [19].

### 2.8. Statistical Analysis

All statistical analyses were performed using the SPSS 25.0 software program (IBM Corp., Armonk, NY, USA). The shear strength, Vickers hardness, mass before and after wear, and wear volume loss data were analyzed with one-way analysis of variance (ANOVA), followed by a post hoc Tukey’s test. The significance level was set at 0.05 (*p* = 0.05).

## 3. Results

### 3.1. Elemental Composition Analysis

A representative XRF spectrum and EDS image and spectrum are shown in Figure 5 and Figure 6. For XRF, the elements of Al, Si, Cr, Fe and Mo appeared in common in all groups. The DM group presented the highest Fe content of 83.17 at%, and the Fe content values were high and in the order of 82.85 at% in the AM group, 81.48 at% in the EM group and 80.21 at% in the OM group. For EDS, the elements of C, Cr, Fe, Si and O appeared in all groups. And the values of Fe and Cr were similar in XRF and EDS.

### 3.2. Shear Strength

The means and standard deviations of the shear strength for each group are presented in Figure 7A. The shear strength of the DM group was significantly higher than that of the other groups (*p* < 0.05). The OM groups have shear strengths of 80.61 ± 8.34 MPa, significantly lower (*p* < 0.05) than that of the DM group (248.07 ± 8.09 MPa). One-way analysis of variance showed a statistically significant difference among all groups (*p* < 0.05).

### 3.3. Vickers Hardness

The values obtained for the Vickers hardness of the commercial stainless-steel ultrasonic scaler tip are shown in Figure 7B. The Vickers hardness values of the samples ranged from 543.19 to 604.51. There was no significant difference in the mean Vickers hardness values between the AM group and the OM group (*p* > 0.05). The Vickers hardness value increased in the order of DM, EM, OM, and AM, and that of the DM group was significantly higher than that of the other groups (*p* < 0.05).

### 3.4. Wear Degree by Mass and Volume

The image analyzer and SEM images of each commercial stainless-steel ultrasonic scaler tip before and after the wear test are shown in Figure 8. The analyzed surfaces showed clear differences before and after wear in terms of form. In the image analyzer image, the commercial stainless-steel ultrasonic scaler tips of all groups showed a round shape at the end of the tip before the wear test, but the rounded tip after the wear test was worn and deformed linearly (Figure 8; upper row). In the SEM image, all commercial stainless-steel ultrasonic scaler tips showed a smooth surface before the wear test but showed rounded wear of the surface after the wear test (Figure 8; lower row).

The degree of wear by mass and volume (*n* = 8) of commercial stainless-steel ultrasonic scaler tips are given in Table 2 and Figure 9. The amount of wear was between 0.04 ± 0.07 mg and 0.13 ± 0.12 mg. And there was no significant difference among all groups (*p* > 0.05).

The highest wear volume loss values were found in the AM group (21.96 ± 0.67 µm^3^), and the lowest values were found in the DM group (5.93 ± 0.87 µm^3^). Statistical analysis revealed a difference between the AM group and the other groups (*p* < 0.05). These results show that the DM group has the highest wear resistance and that the AM group has the lowest wear resistance.

### 3.5. Grain Size

Figure 10 presents OM images of the polished and etched stainless-steel ultrasonic scaler tip. The grain size appeared to vary depending on manufacturer. After measurement, DM had a higher mean grain size of 30.16 ± 1.68 µm compared to the grain sizes of AM, EM and OM, which were 26.91 ± 3.12 µm, 12.74 ± 1.91 µm and 9.96 ± 2.39 µm, respectively.

## 4. Discussion

Ultrasonic scalers have various advantages over hand instruments, and they are continuously used to remove plaque in the mouth to create a healthy oral hygiene for implant success and periodontal disease prevention [4,7,20]. During normal clinical usage, frequent or prolonged use of scaler tips may cause wear. As the scaler tip wears, it reduces dental calculus removal efficiency and causes damage to teeth [14,15,21]. The scaler tip is no longer efficient, and the tip must be replaced. Scaler tips can be made from a variety of materials, and there have been several studies on clinical use [12,17,22]. Despite many efforts, the least worn material, the stainless-steel scaler tip, is the most commonly used. However, few studies have compared various commercial stainless-steel ultrasonic scale tips. Therefore, in the present study, the element composition and grain size of commercial stainless-steel ultrasonic scaler tips, and the differences in mechanical properties and wear resistance according to this were evaluated.

The strength of a metal is one of the most important properties of metals and plays an important role in designing various components and structures [23]. Figure 7A shows that the shear strength increased in the order DM, AM, EM and OM (*p* < 0.05). In the case of XRF, the content of Fe was in the order DM > AM > EM > OM, which is the same as the strength order. This appears that Fe content affect the shear strength. However, few papers have shown the effect of Fe on stainless steel, and this part may require further studies in the future. According to the work of Hall [24] and Petch [25], it is well known that the strength of a metal depends on grain size. In this study, it was confirmed that the shear strength increased as the grain size decreased. Wengo et al. indicated that the tensile strength and yield strength increase with decreasing the grain size of 316L stainless steel [23], and Chen et al. reported that at a given wire diameter of Ag microwires, as the grain size increases, both the tensile elongation and the ultimate tensile strength decrease [26].

Hardness and wear are the most common mechanical properties of dental materials and are among the most important factors controlling the life of a material [27,28]. Thus, a Vickers hardness tester was used to compare the surface Vickers hardness of the stainless-steel ultrasonic scaler tips. Hardness and wear resistance are known to be correlated [29,30]. The Vickers hardness of stainless-steel ultrasonic scaler tip samples is shown in Figure 7B. The DM group had significantly higher surface Vickers hardness values than the other groups (*p* < 0.05). These results are the same for the wear tests, indicating that the wear resistance increased due to the increase in hardness. The results of increased wear resistance with increasing hardness were the same as those of other studies. Rong et al. showed a consistent relationship in wear testing: the harder the high-C CoCrMo Stellite alloy was, the higher its resistance to alloy wear [31], and Bressan et al. studied the effect of hardness on wear resistance through pin-on-disc testing, indicating that the 17-4 PH stainless-steel disc wear rate increased as the pin hardness decreased [32]. Jeong et al. reported that the wear resistance increased significantly due to the increase in hardness, and this hardness increased as the grain size decreased. This also showed that reduced grain size resulted in higher hardness and increased wear resistance [28]. However, this study did not show the result of increasing the Vickers hardness as the grain size decreased. Hardness is significantly influenced not only by particle size but also by the heat treatment process, time and quenching conditions [33,34,35]. The samples used in this study were subjected to different heat treatment processes depending on the manufacturer, and this is considered to have an effect on the Vickers hardness value. Wear may be mild or severe depending on variables such as pressure, contact temperature and material hardness [32], but in this study, the relationship between Vickers hardness and abrasion resistance was significant because the experiment was conducted under the same conditions for the other variables except Vickers hardness.

Based on the above results, the first null hypothesis, that there are no significant differences in mechanical properties and wear resistance depending on the element composition of the commercial stainless-steel ultrasonic scaler tips, was partially accepted. The shear strength of commercial stainless-steel ultrasonic scaler tips was the highest in DM groups, followed by AM, EM, and OM, which showed that the higher the Fe content, the higher the shear strength. The Vickers hardness and wear resistance of commercial stainless-steel ultrasonic scaler tips were not shown to differ depending on the composition of the elements. The second null hypothesis, that there are no significant differences in mechanical properties and wear resistance depending on the grain size of the commercial stainless-steel ultrasonic scaler tips, was partially accepted. The shear strength of commercial stainless-steel ultrasonic scaler tips was higher in order of DM, AM, EM and OM, which showed that the smaller the grain size, the higher the shear strength. The Vickers hardness and wear resistance of commercial stainless-steel ultrasonic scaler tips were not shown to differ depending on the grain size. Although hardness and wear are affected by element composition and grain size, the samples used in this study are considered to have produced the above results because the heat treatment process, time and quilting conditions which varies from manufacturer to manufacturer. This is the limitation of this study, and further research is needed on how it affects metals with the same conditions.

In this study, clinical wear conditions were reproduced using maximum conditions such as a scaler tip of 90 degrees, maximum power, and irrigation. Research in the laboratory is not able to reproduce and satisfy all oral environmental conditions. However, this is related to clinical performance predictions. This study also investigated the grain size of commercial stainless-steel ultrasonic scaler tips, but did not compare the structure. Further research is necessary to investigate the properties of ultrasonic scaler tips under standard clinical conditions and the structures characteristics using XRD.

## 5. Conclusions

Despite the limitations of this study, the DM group had the best shear strength, Vickers hardness, and wear resistance among the commercial stainless-steel ultrasonic scaler tips. As a result, μXRF and SEM/EDS element composition results and observations of grain size with optical microscopes concluded that the Fe content and grain size of stainless steel had a significant effect on shear strength. In addition, stainless-steel ultrasonic scaler tips with higher Vickers hardness values showed greater wear resistance, which would be an important wear characteristic to be checked by clinicians. This study could also be a useful reference for studies of the shear strength, Vickers hardness, and wear resistance of stainless-steel ultrasonic scaler tips.

## Figures and Tables

**Figure 1 materials-14-00827-f001:**
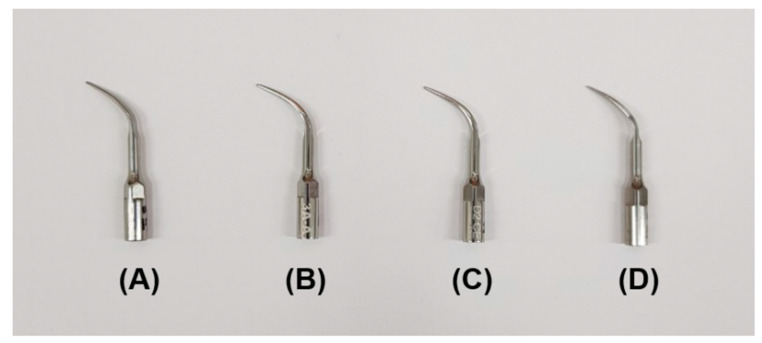
Four commercial stainless steel ultrasonic scaler tips used in the study and produced by (**A**) Electro Medical Systems, (**B**) 3A MEDES, (**C**) DMETEC and (**D**) OSUNG MND.

**Figure 2 materials-14-00827-f002:**
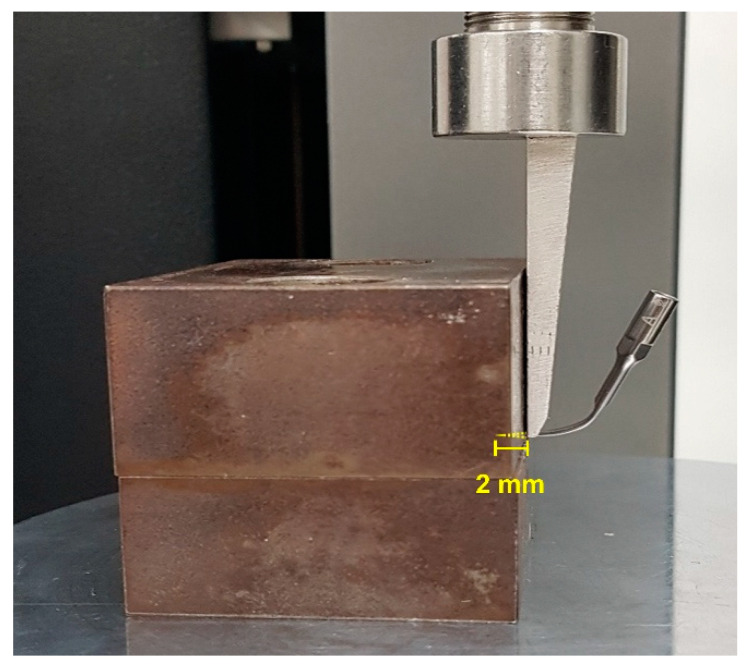
Image of the metal block holding the stainless-steel ultrasonic scaler tip firmly in place and the metal block contact surface parallel to the jig.

**Figure 3 materials-14-00827-f003:**
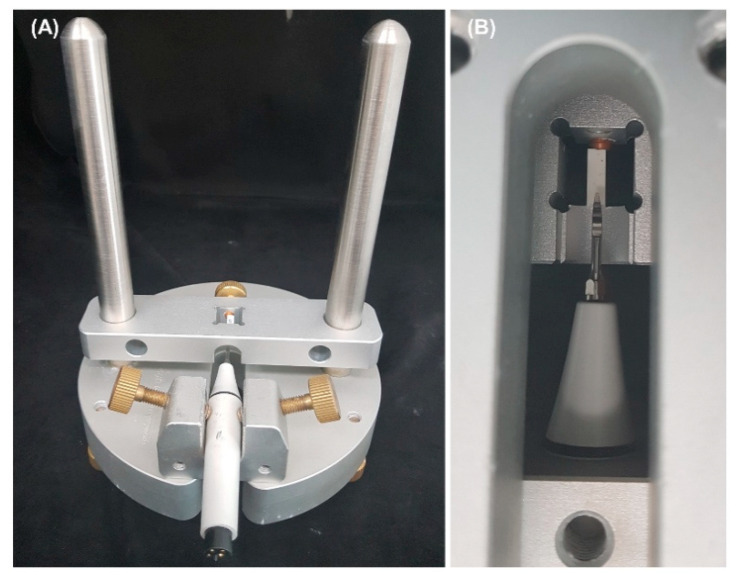
(**A**) Custom jig used to test the wear of a stainless-steel ultrasonic scaler tip. (**B**) Shape of a stainless-steel ultrasonic scaler tip during the wear test, as seen from the bottom.

**Figure 4 materials-14-00827-f004:**
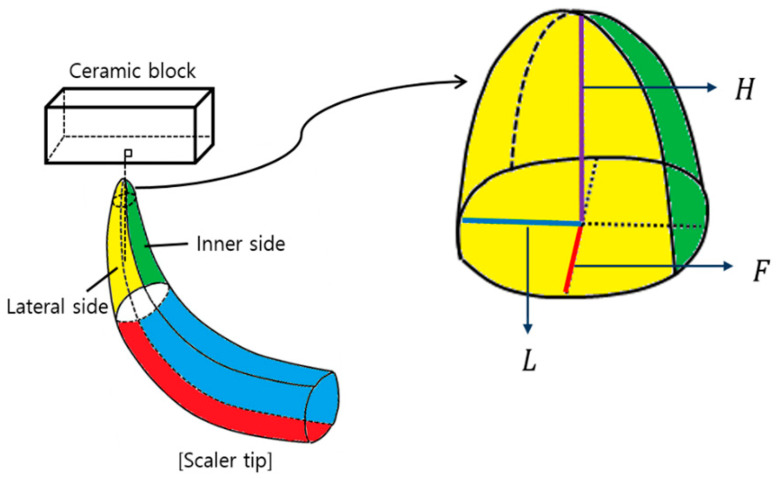
Shape of the stainless-steel ultrasonic scaler tip used for the calculation of the stainless-steel ultrasonic scaler tip wear volume loss.

**Figure 5 materials-14-00827-f005:**
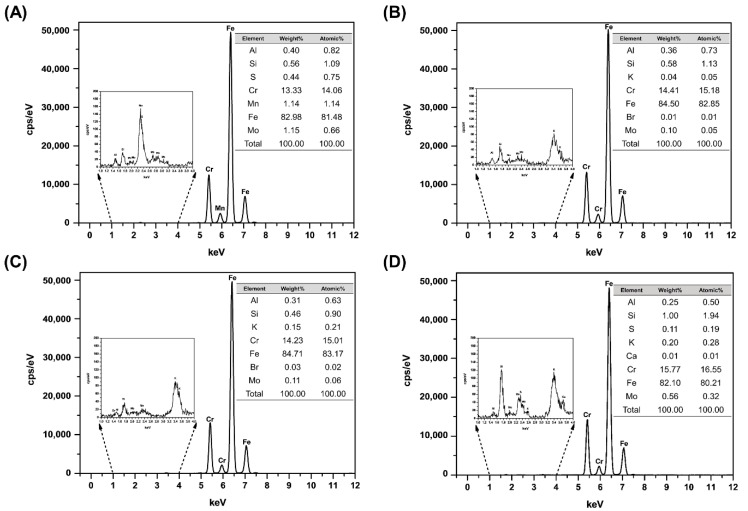
XRF spectra of the (**A**) EM (**B**) AM (**C**) DM and (**D**) OM groups.

**Figure 6 materials-14-00827-f006:**
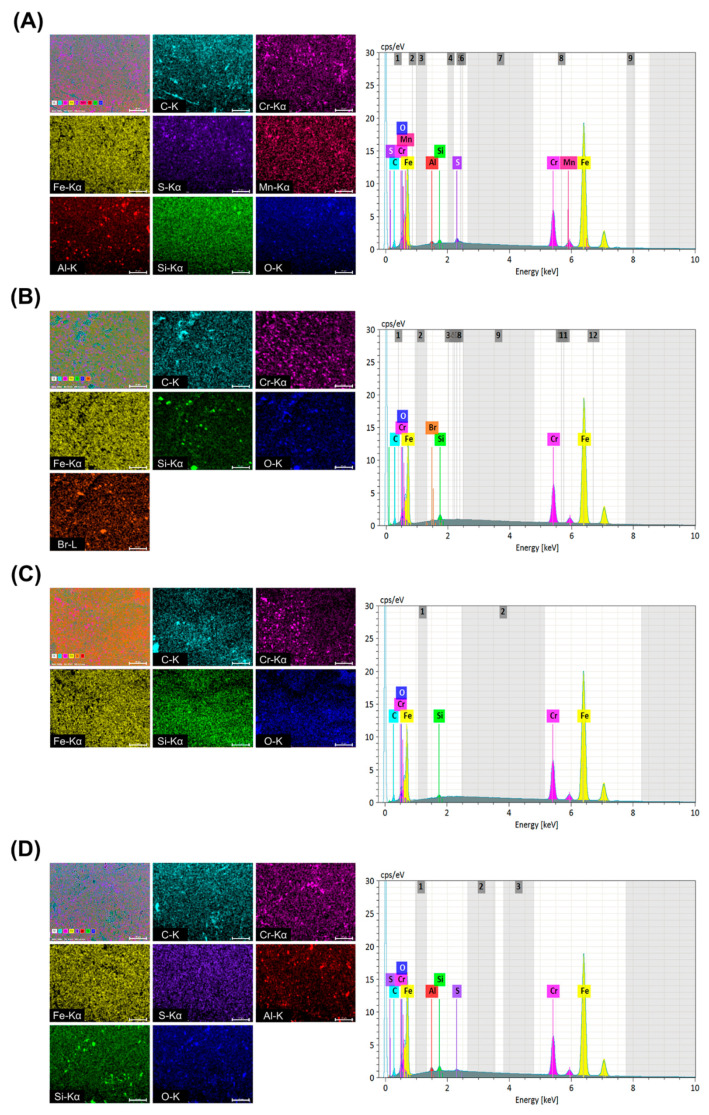
EDS elemental mapping images and spectrum of the (**A**) EM (**B**) AM (**C**) DM and (**D**) OM groups.

**Figure 7 materials-14-00827-f007:**
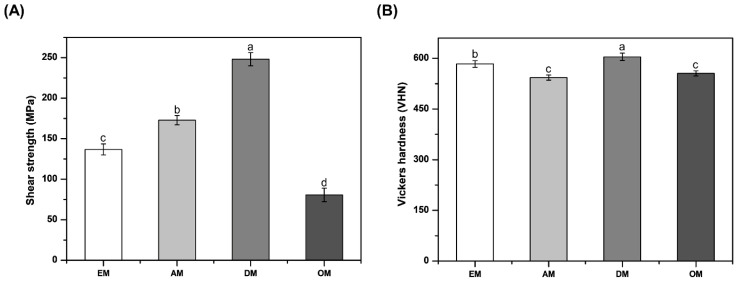
Comparison of shear strength (**A**) and Vickers hardness (**B**) for each group. Differences in lowercase alphabetical letters above the bar graphs indicate significant differences among the groups (*p* < 0.05).

**Figure 8 materials-14-00827-f008:**
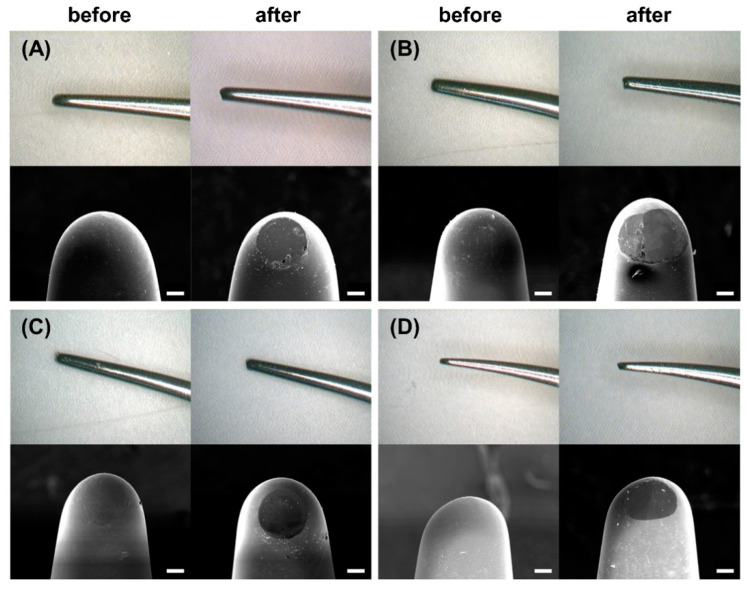
Representative images analyzer (upper row) and SEM images (lower row) obtained for each group of surfaces before and after the wear test: (**A**) EM, (**B**) AM, (**C**), DM and (**D**) OM. Scale bar is 1 µm.

**Figure 9 materials-14-00827-f009:**
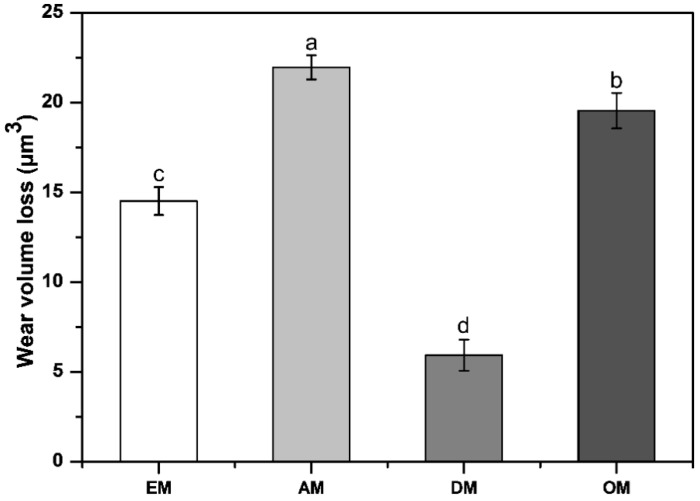
Comparison of wear volume loss for each group after the wear test. Differences in lowercase alphabetical letters above the bar graph indicate significant differences among the groups (*p* < 0.05).

**Figure 10 materials-14-00827-f010:**
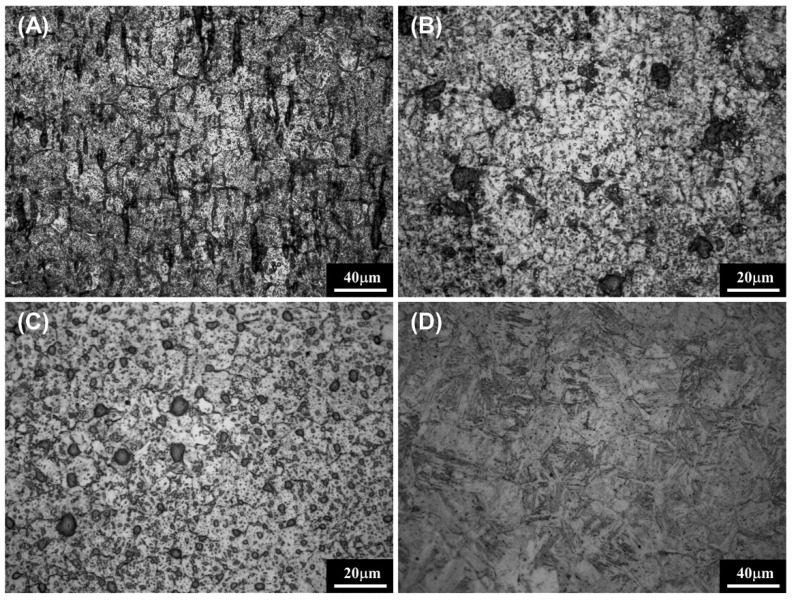
Microstructures of (**A**) EM, (**B**) AM, (**C**) DM and (**D**) OM observed by optical microscopy. Scale bar is 40 µm (**A**,**D**) and 20 µm (**B**,**C**).

**Table 1 materials-14-00827-t001:** The four commercial stainless-steel ultrasonic scaler tips used in this study.

Group	Manufacturer	Product Name	Location	Materials	Purpose
EM	Electro Medical Systems	A Type	Nyon, Switzerland	Stainless steel	Full mouth/ Supragingival
AM	3A MEDES	A Korea	Gyeonggi-do, Korea	Stainless steel	Full mouth/ Supragingival
DM	Dmetec	D2	Gyeonggi-do, Korea	Stainless steel	Full mouth/ Supragingival
OM	OSUNG MND	USEA	Gyeonggi-do, Korea	Stainless steel	Full mouth/ Supragingival

**Table 2 materials-14-00827-t002:** Comparison of wear degree by mass for each group.

Group	Before Mass (mg)	After Mass (mg)	Amount of Wear (mg)
EM	813.69 ± 3.98	813.59 ± 4.01	0.10 ± 0.09 *
AM	822.43 ± 6.93	822.39 ± 6.90	0.04 ± 0.07 *
DM	807.75 ± 2.91	807.70 ± 2.88	0.05 ± 0.08 *
OM	815.50 ± 2.92	815.38 ± 2.95	0.13 ± 0.12 *

SD—standard deviation; There were no significant differences in values marked with * (*p* > 0.05).

## Data Availability

Data is contained within the article.

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
