# Peer review of "Mechanical Properties and Wear Resistance of Commercial Stainless Steel Used in Dental Instruments"

_materials, 2021, doi:10.3390/ma14040827_

Round 1
Reviewer 1 Report
The paper is devoted to a comparative study of strength, hardness and wear of commercial stainless steel ultrasonic scaler tips from different manufacturers. Although the authors used many different methods and the results obtained may undoubtedly be of practical interest, the study, in my opinion, was carried out superficially without an adequate discussion of the results. I cannot recommend its publication.
Several issues will be required before this paper can be considered for publication:
1) Why did the authors suggest null hypotheses which, moreover, it is not clear what they are based on?
2) How did the authors measure Vickers hardness? On the surface of the samples or at a depth ? This information must be provided.
3) How did the authors perform metallographic analysis? Was it a longitudinal or transverse section? This information must be provided.
4) What is the measurement accuracy of the XRF method? For some reason the authors focus on the impurities of K and Br, the concentration of which is close to zero. At the same time, the authors argue that the manganese content varied slightly between the group, although based on Figure 5, manganese was present only in the EM samples. And the authors ignore the different concentration of molybdenum between the groups.
5) If, according to the authors, there was no significant difference in the degree of wear by weight between all groups, then why do the authors argue that the degree of wear by weight was maximum in the OM group, and the least in the AM group?
6) The authors report the average grain size, but do not report the deviation. By the way, the samples apparently differed not only in the grain size, but also in the type of structure. In this case, it makes no sense to associate a change in properties only with a change in grain size. The authors do not discuss a different type of structure of samples between different groups.
7) When discussing the effect of chemical elements on strength, the authors compare completely different materials (aluminum alloys, nickel alloys, etc.), which is completely incorrect.
Others:
- It is recommended to shorten the title of the manuscript, for example: Shear strength, Vickers hardness and wear resistance of commercial stainless steel ultrasonic scaler tips.
- There is no caption for Table 1.
- The values ​​of strength, microhardness and grain size in the text of the manuscript should be rounded off to at least tenths.
Author Response
Reviewer #1 |
|
Comment 1 |
Why did the authors suggest null hypotheses which, moreover, it is not clear what they are based on? |
Response to Comment 1 |
Thank you for kind help on reviewing this article. Your valuable opinions have assisted me to gain new knowledge related to this field and to have a clearer understanding of the limitations in this research. According to your suggestion, we have now revised hypothesis of the manuscript with addition of background information as follows: 1.Introduction However, there are no studies investigating the element composition and grain size of commercial stainless steel ultrasonic scaler tips and to compare the differences in me-chanical properties and wear resistance according to this. Therefore, the aim of this study was to investigate the element composition and grain size of commercial stainless steel ultrasonic scaler tips and to compare the differneces in mechanical properties and wear resistance according to this. The null hypotheses were as follows: (1) there would be no difference in mechanical properties and wear resistance depending on the element composition of the commercial stainless steel ultrasonic scaler tips (2) there would be no difference in mechanical properties and wear resistance de-pending on the grain size of the commercial stainless steel ultrasonic scaler tips. |
|
|
Comment 2 |
How did the authors measure Vickers hardness? On the surface of the samples or at a depth? This information must be provided. |
Response to Comment 2 |
Sorry for the lack of information, and above information are now added in the methods sections. Ten samples were prepared for each stainless steel ultrasonic scaler tip group. When measuring the Vickers hardness of the scalar tip, the end of the scalar tip was fixed flat with the bottom, and was measured at a surface of 2 mm at the end of the tip. The Vickers hardness of the sample was determined with a micro Vickers hardness tester (DMH-2, Matsuzawa Seiki Co., Tokyo, Japan) using a Vickers diamond indenter and a 500 g load applied for 10 s, and both indentation diagonals (mm) were measured. |
|
|
Comment 3 |
How did the authors perform metallographic analysis? Was it a longitudinal or transverse section? This information must be provided. |
Response to Comment 3 |
Again, sorry for the lack of information. We’ve added details as follows: The microstructure surface samples were mounted on epoxy resin so that the work-ing end of the scaler tip was parallel to the floor, then mechanically polished se-quentially using 800-grit, 1000-grit, 1500-grit, 2000-grit SiC papers and standard colloidal silica suspension. Then, the well-polished samples were etched for 2-3 min using a solu-tion of 30 ml hydrochloric acid, 30 ml nitric acid and 40 ml distilled water to reveal the grain boundaries [17]. After that, optical images at 500x magnification and 1000x magni-fication were captured with an optical microscope (OM, JE clipse LV150N, Nikon, Tokyo, Tokyo). The average grain size was determined based on the OM images using ImageJ software. |
|
|
Comment 4 |
What is the measurement accuracy of the XRF method? For some reason the authors focus on the impurities of K and Br, the concentration of which is close to zero. At the same time, the authors argue that the manganese content varied slightly between the group, although based on Figure 5, manganese was present only in the EM samples. And the authors ignore the different concentration of molybdenum between the groups. |
Response to Comment 4 |
According to your suggestion, we have revised the manuscript as follows with more details: 2. Materials and methods The elemental compositions of the stainless steel ultrasonic scaler tips were deter-mined using micro-X-ray fluorescence spectrometry (μXRF) analysis (M4 TORNADO, Bruker, MA, USA) with a rhodium filament operating at 50 kV and 40 mA and with a spot size of 25 µm. The system comprises a Rh anode X-ray tube and an XFlash detector (energy resolution 145 eV for Mn-Kα and 30 mm2 active). Two sites were measured at random for each sample, and the mean values were obtained and compared. 3. Results A representative XRF spectrum and EDS image and spectrum are shown in Figure 5 and 6. For XRF, the elements of Al, Si, Cr, Fe and Mo appeared in common in all groups. The DM group presented the highest Fe content of 83.17 at%, and the Fe content values were high and in the order of 82.85 at% in the AM group, 81.48 at% in the EM group and 80.21 at% in the OM group. For EDS, the elements of C, Cr, Fe, Si and O appeared in all groups. And the values of Fe and Cr were similar in XRF and EDS. |
|
|
Comment 5 |
If, according to the authors, there was no significant difference in the degree of wear by weight between all groups, then why do the authors argue that the degree of wear by weight was maximum in the OM group, and the least in the AM group? |
Response to Comment 5 |
Sorry for the confusion in our manuscript. The related sentences were modified in the results sections for the clarification. The degree of wear by weight and volume (n=8) of commercially available stainless steel ultrasonic scaler tips are given in Table 2 and Figure 8. The amount of wear was between 0.04 ± 0.07 mg and 0.13 ± 0.12 mg. And there was no significant difference among all groups (p>0.05). |
|
|
Comment 6 |
The authors report the average grain size, but do not report the deviation. By the way, the samples apparently differed not only in the grain size, but also in the type of structure. In this case, it makes no sense to associate a change in properties only with a change in grain size. The authors do not discuss a different type of structure of samples between different groups. |
Response to Comment 6 |
According to your suggestion, we have revised the manuscript as follows with deviation values along with average size. Also, type of structure were considered in discussion: 3. Results Figure 9 presents OM images of the polished and etched stainless steel ultrasonic scaler tip. The grain size appeared to vary depending on manufacturer. After measurement, DM had a higher mean grain size of 30.16 ± 1.68 µm compared to the grain sizes of AM, EM and OM, which were 26.91 ± 3.12 µm, 12.74 ± 1.91 µm and 9.96 ± 2.39 µm, respectively. 4. Discussion In this study, clinical wear conditions were reproduced using maximum conditions such as a scaler tip of 90 degrees, maximum power, and irrigation. Research in the labor-atory is not able to reproduce and satisfy all oral environmental conditions. However, this is related to clinical performance predictions. This study also investigated the grain size of commercial stainless steel ultrasonic scaler tips, but did not compare the structure. Further research is necessary to investigate the properties of ultrasonic scaler tips under standard clinical conditions and the structures characteristics using XRD. |
|
|
Comment 7 |
When discussing the effect of chemical elements on strength, the authors compare completely different materials (aluminum alloys, nickel alloys, etc.), which is completely incorrect. |
Response to Comment 7 |
Sorry for the confusion in such references. According to your suggestion, the related sentences were modified in the discussion sections. The strength of a metal is one of the most important properties of metals and plays an important role in designing various components and structures [23]. Figure 6A shows that the shear strength increased in the order DM, AM, EM and OM (p <0.05). In the case of XRF, the content of Fe was in the order DM>AM>EM>OM, which is the same as the strength order. The influences of Fe content on mechanical properties are similar to the results of previous work. Lorella et al. reported that high Fe and Mn contents increased the tensile strength of Al-Si10-Cu2 alloys [24], and Wang et al. proved that the more Fe that was added to Ni-Mn-Ga alloys, the better the fracture toughness [25]. Additionally, Yoshinori et al. revealed that the tensile strength of Ti-6Al-4V increases as the Fe or Cr content increases and that the effect of Fe addition on strengthening Ti-6Al-4V was greater than that of Cr addition [26]. However, the materials considered in the previous paper were Al Alloy, Ni Alloy and Ti, which could not be fully applied to the stainless steel used in this paper, but the effect on Fe in metals could be considered. |
|
|
Minor Comment |
- It is recommended to shorten the title of the manuscript, for example: Shear strength Vickers hardness and wear resistance of commercial stainless steel ultrasonic saler tips. - There is no caption for Table 1. - The values of strength, microhardness and grain size in the text of the manuscript should be rounded off to at least tenths. |
Response to Comment |
Thank you for your comment and we have now revised title and other details in accordance to your comments. |
|
|
Reviewer 2 Report
The manuscript covers a very interesting topic concerning the clinical applications of ultrasonic scaler tips. I recommend the work to be approved after answering the following questions: Please clearly indicate the novelty of this study compared to other methods, coatings. Authors should better emphasize the merits of their method than others. Please add dates with SEM / EDS. This study is supported by examination of the structures and morphology of commercially available stainless steel ultrasonic scaler tips. Please explain the conclusions of the article in more detail.
Author Response
Reviewer #2 |
|
Comment 1 |
Please clearly indicate the novelty of this study compared to other methods, coatings. Authors should better emphasize the merits of their method than others. |
Response to Comment 1 |
Thank you for your thoughtful comments on the manuscript. We now have added details on the novelty of the research along with methods as follows: Therefore, the aim of this study was to investigate the element composition and grain size of commercial stainless steel ultrasonic scaler tips and to compare the differneces in mechanical properties and wear resistance according to this. The null hypotheses were as follows: (1) there would be no difference in mechanical properties and wear resistance depending on the element composition of the commercial stainless steel ultrasonic scaler tips (2) there would be no difference in mechanical properties and wear resistance de-pending on the grain size of the commercial stainless steel ultrasonic scaler tips. |
|
|
Comment 2 |
Please add dates with SEM / EDS. |
Response to Comment 2 |
According to your suggestion, SEM/EDS data were added as Figure 6 with additional details in Materials and Methods, and Results. 2. Materials and methods The elemental compositions of the stainless steel ultrasonic scaler tips were deter-mined using micro-X-ray fluorescence spectrometry (μXRF) analysis (M4 TORNADO, Bruker, MA, USA) with a rhodium filament operating at 50 kV and 40 mA and with a spot size of 25 µm. The system comprises a Rh anode X-ray tube and an XFlash detector (energy resolution 145 eV for Mn-Kα and 30 mm2 active). Two sites were measured at random for each sample, and the mean values were obtained and compared. Also, the field-emission scanning electron microscopy (FE-SEM; MERLIN, ZEISS, Ltd., Oberkochen, Germany) equipped with energy-dispersive X-ray spectroscopy (EDS) were used to deter-mine the elemental compositions. 3. Results A representative XRF spectrum and EDS image and spectrum are shown in Figure 5 and 6. For XRF, the elements of Al, Si, Cr, Fe and Mo appeared in common in all groups. The DM group presented the highest Fe content of 83.17 at%, and the Fe content values were high and in the order of 82.85 at% in the AM group, 81.48 at% in the EM group and 80.21 at% in the OM group. For EDS, the elements of C, Cr, Fe, Si and O appeared in all groups. And the values of Fe and Cr were similar in XRF and EDS. |
|
|
Comment 3 |
This study is supported by examination of the structures and morphology of commercially available stainless steel ultrasonic scaler tips. Please explain the conclusions of the article in more detail. |
Response to Comment 3 |
Sorry for lack of information and in accordance to your suggestion, we’ve added details in conclusion as follows: Despite the limitations of this study, the DM group had the best shear strength, Vickers hardness, and wear resistance among the commercial stainless steel ultrasonic scaler tips. As a result, μXRF and SEM/EDS element composition results and observations of grain size with optical microscopes concluded that the Fe content and grain size of stain-less steel had a significant effect on shear strength. In addition, stainless steel ultrasonic scaler tips with higher Vickers hardness values showed greater wear resistance, which would be an important wear characteristic to be checked by clinicians. This study could also be a useful reference for studies of the shear strength, Vickers hardness and wear resistance of stainless steel ultrasonic scaler tips. |
|
|
Reviewer 3 Report
In the reviewed article the issue of the influence of the properties of acid-resistant steel materials on the wear of scaler tips under laboratory conditions was analyzed. The discussed research problem is of great importance in terms of cognitive and practical application. Knowing what conditions the material should meet to ensure adequate durability and accuracy of the process is always a very important issue not only in dentistry, but also in all technical sciences.
The overall assessment of the work is positive. Work arranged correctly in a logical whole, good research equipment, correct methodologies for their implementation. During a detailed analysis of the article, we encounter numerous inaccuracies and understatements. The main ones are.
- Specifying the title - unnecessary emphasis on the individual characteristics of the surface layer, such as strength, hardness. The values ​​of these characteristics make up the properties of the surface varnish and it is they who determine the course of wear.
- The purpose of the work is included in the summary, why is it not at work?
- The authors freely choose the terms of hardness and microhardness, and there is a fundamental difference between them. There is no standard for determining the hardness and the associated different determination in the text and in the drawing.
- The article uses non-SI units, which in my opinion should be improved.
- The authors mention the manufacturers of the equipment used. Isn't that an advertisement?
- The balance is a device and serves to measure the mass, therefore some of the wording are unacceptable, eg the degree of wear by weight.
- There are no clearly formulated hypotheses in the methodology. They would facilitate the analysis of the results.
- The analysis of the results needs to be refined. There are too many generalities that are obvious to readers. On the other hand, a more extensive analysis is required to justify the obtained results of wear in the context of the properties of the surface layer (chemical composition, grain size for hardness and strength, and then for wear). Moreover, the authors use the certainties from literature (line 283) during their own research. The question arises in what aspect are the obtained results original in relation to those already published?
- Conclusions should be written in relation to a clearly stated goal. After introducing the presented corrections and marked in the text, the paper should be re-reviewed in order to evaluate the introduced corrections.

Author Response
Reviewer #3 |
|
Comment 1 |
Specifying the title - unnecessary emphasis on the individual characteristics of the surface layer, such as strength, hardness. The values of these characteristics make up the properties of the surface varnish and it is they who determine the course of wear. |
Response to Comment 1 |
Thank you for your valuable advice and thoughtful comments on the manuscript. According to your suggestion, we have now modified the title as; Mechanical properties and wear resistance of commercial stainless steel ultrasonic scaler tips |
|
|
Comment 2 |
The purpose of the work is included in the summary, why is it not at work? |
Response to Comment 2 |
Sorry for the confusion and lack of information. We now have added more details in summary of Introduction as well as modifying purpose and hypothesis of this study to clarify the purpose as follows: Therefore, the aim of this study was to investigate the element composition and grain size of commercial stainless steel ultrasonic scaler tips and to compare the differneces in mechanical properties and wear resistance according to this. The null hypotheses were as follows: (1) there would be no difference in mechanical properties and wear resistance depending on the element composition of the commercial stainless steel ultrasonic scaler tips (2) there would be no difference in mechanical properties and wear resistance de-pending on the grain size of the commercial stainless steel ultrasonic scaler tips. |
|
|
Comment 3 |
The authors freely choose the terms of hardness and microhardness, and there is a fundamental difference between them. There is no standard for determining the hardness and the associated different determination in the text and in the drawing. |
Response to Comment 3 |
According to your suggestion, we have revised the all manuscript to one appropriate term. |
|
|
Comment 4 |
The article uses non-SI units, which in my opinion should be improved. |
Response to Comment 4 |
Thank you for your comments. We have now modified units into SI-units as possible, except for units that are commonly used. |
|
|
Comment 5 |
The authors mention the manufacturers of the equipment used. Isn't that an advertisement? |
Response to Comment 5 |
All of manufacturer were stated in accordance to guidance for manuscript writing of Materials. We have no conflict of interest with manufacturers and if required by editorial team, we are happy to blind them. |
|
|
Comment 6 |
The balance is a device and serves to measure the mass, therefore some of the wording are unacceptable, eg the degree of wear by weight. |
Response to Comment 6 |
According to your suggestion, the related sentences were modified and added in the methods and results sections. 2.6. Materials and methods Two methods of calculating the wear by weight mass and volume were used in this study. To evaluate the weight mass change due to the wear test, the weight mass of the stainless steel ultrasonic scaler tip was measured using an analytical balance (Sartorius analytic A120S, Sartorius, Göttingen, Germany). The wear degree by weight mass of all samples was calculated by measuring the weight mass before and after the wear test. 3. Results The degree of wear by weight mass and volume (n=8) of commercially available stainless steel ultrasonic scaler tips are given in Table 2 and Figure 8. The amount of wear was be-tween 0.04 ± 0.07 mg and 0.13 ± 0.12 mg. The wear amount was the highest in the OM group, and the wear amount was the smallest in the AM group. However, And there was no significant difference among all groups (p>0.05). |
|
|
Comment 7 |
There are no clearly formulated hypotheses in the methodology. They would facilitate the analysis of the results. |
Response to Comment 7 |
Sorry for the lack of information and we now have modified hypotheses of the research as follows: Therefore, the aim of this study was to investigate the element composition and grain size of commercial stainless steel ultrasonic scaler tips and to compare the differneces in mechanical properties and wear resistance according to this. The null hypotheses were as follows: (1) there would be no difference in mechanical properties and wear resistance depending on the element composition of the commercial stainless steel ultrasonic scaler tips (2) there would be no difference in mechanical properties and wear resistance de-pending on the grain size of the commercial stainless steel ultrasonic scaler tips. |
|
|
Comment 8 |
The analysis of the results needs to be refined. There are too many generalities that are obvious to readers. On the other hand, a more extensive analysis is required to justify the obtained results of wear in the context of the properties of the surface layer (chemical composition, grain size for hardness and strength, and then for wear). Moreover, the authors use the certainties from literature (line 283) during their own research. The question arises in what aspect are the obtained results original in relation to those already published? |
Response to Comment 8 |
According to your suggestion, the SEM/EDS analysis were added, and limitations on the literature have been specified as follows: 2. Materials and methods The elemental compositions of the stainless steel ultrasonic scaler tips were deter-mined using micro-X-ray fluorescence spectrometry (μXRF) analysis (M4 TORNADO, Bruker, MA, USA) with a rhodium filament operating at 50 kV and 40 mA and with a spot size of 25 µm. The system comprises a Rh anode X-ray tube and an XFlash detector (energy resolution 145 eV for Mn-Kα and 30 mm2 active). Two sites were measured at random for each sample, and the mean values were obtained and compared. Also, the field-emission scanning electron microscopy (FE-SEM; MERLIN, ZEISS, Ltd., Oberkochen, Germany) equipped with energy-dispersive X-ray spectroscopy (EDS) were used to deter-mine the elemental compositions. |
|
|
Comment 9 |
Conclusions should be written in relation to a clearly stated goal. After introducing the presented corrections and marked in the text, the paper should be re-reviewed in order to evaluate the introduced corrections. |
Response to Comment 9 |
According to your suggestion, we now have revised both Introduction and Conclusion of the manuscript as follows: 1. Introduction Therefore, the aim of this study was to investigate the element composition and grain size of commercial stainless steel ultrasonic scaler tips and to compare the differneces in mechanical properties and wear resistance according to this. The null hypotheses were as follows: (1) there would be no difference in mechanical properties and wear resistance depending on the element composition of the commercial stainless steel ultrasonic scaler tips (2) there would be no difference in mechanical properties and wear resistance de-pending on the grain size of the commercial stainless steel ultrasonic scaler tips. 5. Conclusions Despite the limitations of this study, the DM group had the best shear strength, Vickers hardness, and wear resistance among the commercial stainless steel ultrasonic scaler tips. As a result, μXRF and SEM/EDS element composition results and observations of grain size with optical microscopes concluded that the Fe content and grain size of stain-less steel had a significant effect on shear strength. In addition, stainless steel ultrasonic scaler tips with higher Vickers hardness values showed greater wear resistance, which would be an important wear characteristic to be checked by clinicians. This study could also be a useful reference for studies of the shear strength, Vickers hardness and wear resistance of stainless steel ultrasonic scaler tips. |
|
|
Reviewer 4 Report
Unfortunately, I feel that the main content and focus of this work were more about performance testing of commercial products, which makes it does not fit well the scope of the journal. Also, the methods of investigation were relatively simple and not professional for materials. Last but not the least, the conclusions of this manuscript were not conclusive actually. The authors are suggested to add more results and submit this manuscript to dentistry related journals.
Author Response
Reviewer #4 |
|
Comment 1 |
Unfortunately, I feel that the main content and focus of this work were more about performance testing of commercial products, which makes it does not fit well the scope of the journal. Also, the methods of investigation were relatively simple and not professional for materials. Last but not the least, the conclusions of this manuscript were not conclusive actually. The authors are suggested to add more results and submit this manuscript to dentistry related journals. |
Response to Comment 1 |
Thank you for your valuable advice on the manuscript. First, we modified the purpose, hypothesis to indicate suitability of this manuscript related to scope of this journal. Also, we revised the conclusion section as a whole. Finally, we proved that it is a paper related to materials by analyzing SEM/EDS. |
|
|
Round 2
Reviewer 1 Report
Although the authors addressed many issues raised by the reviewer in the revised paper, some are still to be improved further.
I once again urge the authors not to compare steels with aluminum alloys, titanium, etc.
Author Response
Reviewer #1 |
|
Comment 1 |
I once again urge the authors not to compare steels with aluminum alloys, titanium, etc. |
Response to Comment 1 |
Sorry for misunderstanding initial revision comment and thank you for your thoughtful comments on the manuscript. We have now revised discussion details in accordance to your comments, which avoided comparison of steels with aluminium alloys, titanium etc. |
Reviewer 3 Report
Dear Sir,
Thank you very much for your comments. They explained the applied research procedures and allowed for the correct interpretation of the results.
However, I do not agree with the designation of Vickers hardness as Hv. It should be in accordance with applicable HV standards. I leave the decision to the editorial office.
Best regards
Author Response
Reviewer #3 |
|
Comment 1 |
However, I do not agree with the designation of Vickers hardness as Hv. It should be in accordance with applicable HV standards. I leave the decision to the editorial office. |
Response to Comment 1 |
Sorry for the confusion. We now have checked designation with International Standard, ISO 6507-1:2018 Metallic materials – Vickers hardness test. It stated the designation as HV and hence we though it would be acceptable terms, and corrected manuscript accordingly. |
Reviewer 4 Report
I will have reservations on the suitability of publishing this work in Materials. Unfortunately, I cannot see significant improvement in terms of data quality and content in the revised manuscript. More importantly, the materials characterization and mechanical testing involved in this work were not professional. My previous comments still stand. This work is more suitable to dentistry related journals.
Author Response
Reviewer #4 |
|
Comment 1 |
I will have reservations on the suitability of publishing this work in Materials. Unfortunately, I cannot see significant improvement in terms of data quality and content in the revised manuscript. More importantly, the materials characterization and mechanical testing involved in this work were not professional. My previous comments still stand. This work is more suitable to dentistry related journals. |
Response to Comment 1 |
Sorry for the confusion on the scope of this paper. As the scope of Materials would include “all kinds of functional materials including material for dentistry”, we though this would be suitable. We have further modified abstract and title to ensure relevance with this journal. |